# Sensitivity Analysis of Biome-BGC for Gross Primary Production of a Rubber Plantation Ecosystem: A Case Study of Hainan Island, China

**DOI:** 10.3390/ijerph192114068

**Published:** 2022-10-28

**Authors:** Junyi Liu, Zhixiang Wu, Siqi Yang, Chuan Yang

**Affiliations:** 1College of Ecology and Environment, Hainan University, Haikou 570228, China; 2Rubber Research Institute, Chinese Academy of Tropical Agricultural Sciences, Haikou 571101, China; 3Hainan Danzhou Tropical Agro-Ecosystem National Observation and Research Station, Danzhou 571737, China

**Keywords:** carbon flux simulation, Biome-BGC, rubber plantation ecosystem, eFAST sensitivity analysis

## Abstract

Accurate monitoring of forest carbon flux and its long-term response to meteorological factors is important. To accomplish this task, the model parameters need to be optimized with respect to in situ observations. In the present study, the extended Fourier amplitude sensitivity test (eFAST) method was used to optimize the sensitive ecophysiological parameters of the Biome BioGeochemical Cycles model. The model simulation was integrated from 2010 to 2020. The results showed that using the eFAST method quantitatively improved the model output. For instance, the R^2^ increased from 0.53 to 0.72. Moreover, the root-mean-square error was reduced from 1.62 to 1.14 gC·m^−2^·d^−1^. In addition, it was reported that the carbon flux outputs of the model were highly sensitive to various parameters, such as the canopy average specific leaf area and canopy light extinction coefficient. Moreover, long-term meteorological factor analysis showed that rainfall dominated the trend of gross primary production (GPP) of the study area, while extreme temperatures restricted the GPP. In conclusion, the eFAST method can be used in future studies. Furthermore, eFAST could be applied to other biomes in response to different climatic conditions.

## 1. Introduction

Because of fossil fuel combustion and changes in land use and land cover, the increase in the atmospheric CO_2_ concentration has become a common problem in recent years [1]. Forests are important carbon sinks for terrestrial ecosystems, and rational use of the carbon sequestration function of forest ecosystems can effectively alleviate the aggravation of the greenhouse effect [2]. Rubber plantations widely planted in Southeast Asia and tropical China can produce natural rubber, provide timber, and also have a considerable carbon sink capacity [3]. At present, the rubber planting area on Hainan Island has reached 5.43 × 10^5^ hectares, accounting for more than 1/4 of the total forest area of the island [4]. Therefore, rubber plantations may contribute significantly to global carbon sequestration, and it is of great significance to accurately monitor the carbon sink capacity of rubber plantations and their long-term responses to meteorological factors. Currently, methods for monitoring forest carbon sinks include eddy covariance measurements, remote sensing techniques, and model simulation. However, due to differences in specific research objectives, these methods have certain shortcomings [5]. For example, eddy covariance measurements can monitor precise exchanges of CO_2_, water, and energy between forests and the atmosphere on smaller timescales, but are expensive and limited in scope [6]. Remote sensing techniques can provide continuous observations on a larger scale, but the monitoring results are strongly influenced by clouds and rain and have low accuracy [7]. Model simulation can describe the production of plants at different stages and has good scalability, but there are general problems of low simulation accuracy and large simulation uncertainty under different vegetation types and climatic conditions [8]. A common approach today is to combine models with multiple sources of observations to achieve accurate simulations [9].

Process-based models treat ecosystems as an atmosphere–vegetation–soil continuum to describe the dynamics of carbon, nitrogen, and water stocks and fluxes between the different modules [10,11]. Among them, the Biome-BioGeochemical Cycles (Biome-BGC) model is widely used because it is driven by traditional meteorological data [12]. The model has good scalability and can not only simulate the carbon sink changes of biological communities under different environmental factors and human management modes through structural adjustment [13,14], but also can be combined with remote sensing data to explore the spatiotemporal dynamics of vegetation at the regional scale [15,16,17]. However, this model was originally designed for temperate forests, and the phenological differences of different vegetation types were ignored in its structure, so the application of the model in tropical regions has been relatively rare and shown some simulation errors. The model uses more than 40 ecophysiological parameters, introducing high uncertainty in the simulation results [18]. The commonly used model calibration methods include model parameter optimization based on observation data and data assimilation methods [19]. Sensitivity analysis methods can quantify the contribution of parameters to the model output, which can be used to explain the functional pattern of the model and its parameter optimization [20]. The variance-based global sensitivity analysis method can analyze nonlinear models and is widely used in various process models to identify sensitive parameters for different outputs or the coupling effects of different parameters on the outputs [21]. However, the range of the parameter values used for analysis is usually defined by the model itself or the perturbation of a certain value [22,23]; therefore, the influence of different parameter value ranges on the analysis results is still uncertain.

The objective of this study was to optimize the process model using parameter sensitivity analysis methods and parameter optimization methods based on observation data to estimate the long-term carbon flux variability in the study area and explore its responses to meteorological factors. The study used the extended Fourier amplitude sensitivity test (eFAST) method to explore the most sensitive parameters for the carbon flux outputs of Biome-BGC model, and then used the model-independent parameter estimation (PEST) method and the observation data of the eddy covariance system to optimize the model parameters. The 11-year net ecosystem exchange (NEE) and gross primary production (GPP) in the study area were simulated, and their relationships with air temperature and rainfall were analyzed. The study attempted to discuss the following questions: (1) What are the most sensitive ecophysiological parameters for the carbon flux of the Biome-BGC model in rubber plantations? (2) Is Biome-BGC suitable for carbon flux simulation of the rubber plantation in the study area? (3) How does the carbon flux of the rubber plantation in the study area respond to long-term changes in meteorological factors? To answer these questions, the study designed the following numerical experiments: (1) to identify the most sensitive parameters for the model output, as well as their variability for different outputs in different simulation years; (2) to analyze the differences in parameters’ sensitivity to different parameter value ranges; (3) to compare the differences in the model simulation results by choosing different parameters for optimization; and (4) to simulate carbon flux in the study area with optimal parameters and analyze its response to long-term changes in meteorological factors.

## 2. Materials and Methods

### 2.1. Study Area

The observation site—the Hainan Danzhou Tropical Agro-Ecosystem National Observation and Research Station (19°31′47″ N, 109°28′30″ E)—is located in the northwestern part of Hainan Province, P. R. China, and the site is in a gently sloping hilly area with an average altitude of 144 m. The city of Danzhou has a tropical island monsoon climate, with an annual average temperature of 20.5 °C to 28.5 °C. There are obvious dry and rainy seasons in the year, among which May to October is the rainy season and November to April of the following year is the dry season, with an annual average rainfall of 1606~2000 mm. The rainy season begins in July and ends in September, accounting for more than 70% of the annual rainfall. The main vegetation type in the experimental area is rubber trees, including 4 sample plots of different tree ages. The main types of soil in the study area are sandy loam and sandy brick soil, with a thickness of 100 cm [24]. The eddy covariance observation tower is located in the study area, and the near-surface flux observation has been carried out since November 2009. The eddy covariance system outputs water vapor and CO_2_ exchange data for 30 min through the CR3000 data collector. The 30-min data were further filtered and linearly interpolated to obtain daily-scale NEE data for the rubber plantation in the study area from 2010 to 2020 [25,26].

### 2.2. Biome-BGC Model

The Biome-BGC model is a biogeochemical model developed by the Numerical Terradynamic Simulation Group (NTSG) of the University of Montana to simulate the combined cycle of carbon, nitrogen, and water in regional ecosystems. The model was developed by Forest-BGC. After more than 20 years of improvement and version changes, it can simulate the carbon cycle of terrestrial ecosystems with daily steps [27]. Biome-BGC is capable of simulating carbon balance, including GPP, net primary production (NPP), NEE, and ecosystem respiration (Re). The model has been widely used for carbon budget simulation of multiple biomes. This study used version 4.2, provided by the official website (http://www.ntsg.umt.edu/project/biome-bgc.php (accessed on 24 October 2022)).

#### 2.2.1. Model Parameterization

The input data of the model include three parts: site characteristics data, meteorological data, and ecophysiological parameters. Most of the site characteristics data—such as altitude, latitude, and effective soil depth—are obtained through field measurements, and other data—such as soil texture and CO_2_ concentration—are obtained from the relevant literature (Table 1). Meteorological data include daily maximum temperature, minimum temperature, average temperature, rainfall, vapor pressure deficit (VPD), solar radiation, and day length from sunrise to sunset. This study used the daily temperature and rainfall data from 2010 to 2020 provided by the Danzhou Meteorological Observatory and used the MT-CLIM to simulate VPD, solar radiation, and day length. Ecophysiological parameters are model-defined parameters used to characterize the vegetation of a specific biome. The model provides ecophysiological parameters for seven vegetation types, such as evergreen needle-leaved forest, deciduous broad-leaved forest, and shrub. This part of the parameters was also the object of the model sensitivity analysis.

#### 2.2.2. Model Calibration

This study used the PEST and the NEE of the observation site in the year 2010 to calibrate the model’s ecophysiological parameters. PEST is a model-independent parameter estimation method that iterates the model and adjusts the parameters based on the difference between the model output and the measured values, converging the difference to a local optimum to obtain optimized parameters [28]. To run the PEST method, a file (*.tpl) indicating the parameters to be optimized, an instruction file (*.ins) containing observation data, and a control file (*.pst) identifying the range of parameter values and the controlling method for reading and writing are generated. In this study, the ecophysiological parameters (*.epc) were used as the parameters to be optimized, and the NEE of the observation site in 2010 was used for the observed values. Model calibration was achieved by continuously calling the model, comparing the simulated and observed values, and further adjusting the model parameters within a certain parameter value range.

#### 2.2.3. Process

The simulation followed a two-step procedure: First, we used the preindustrial CO_2_ levels and nitrogen deposition to initialize the soil carbon and nitrogen pools of the model, until it reached the equilibrium levels of net ecosystem carbon exchange. Second, we simulated the carbon exchange of the study area from 2010 to 2020. The data for 2010 were used for calibration, while those of 2011–2020 were used for validation.

### 2.3. Sensitivity Analysis Experiment

Sensitivity analysis identifies the sensitive parameters for the model by quantifying the magnitude of the influence of the input parameters on the outputs. Commonly used global sensitivity analysis methods include the Sobol method, Morris method, and Fourier amplitude sensitivity test (FAST) method. The eFAST method adopted in this study combines the advantages of the Sobol method and the FAST method and can obtain fully convergent results with less sampling. The eFAST method is a sensitivity analysis method based on variance decomposition, which treats the model as a function in the following form:(1)$$y=f\left(x_{1},x_{2},\ldots ,x_{k}\right)$$

The Fourier transform decomposes the total variance *V* of the model into the variance *V_i_* contributed by the changes of the individual parameters and the variance *V_i,j, …,k_* contributed by the coupling of multiple parameters. Then, the first-order sensitivity index of the parameter *x_i_* can be defined as follows:(2)$$S_{i}=\frac{V_{i}}{V}$$
and the total sensitivity index can be defined as follows:(3)$$S_{Ti}=\frac{V-V_{-i}}{V}$$
where *V_−i_* is the sum of the variances contributed by all parameters unrelated to the parameter *x_i_* [29].

The model defines 43 ecophysiological parameters, some of which are phenological parameters obtained through field observations, and some of which are weakly related to the carbon outputs and are not involved in the sensitivity analysis. Others have strong dependence on one another, and only representative ones are selected for the analysis. Based on conducting trial tests, only 19 ecophysiological parameters were identified for further sensitivity analysis. The value range of the parameters has a great influence on the sensitivity analysis results, so we selected the maximum range of values suitable for the deciduous broadleaf forests of the study area from the literature in order to fully explore the actual sensitivity of the model parameters (Table 2). To calculate the sensitivity of parameters once using the eFAST method, the model needs to be run *n* × *p* times, where *n* is the number of samples and *p* is the number of parameters. The method considers that the result is valid only when the parameter sampling number is greater than 65. To make the result fully convergent, we took the sampling number *n* as 150. We implemented the eFAST method of model parameters in a Python environment using the SALib package (https://github.com/SALib (accessed on 24 October 2022)).

## 3. Results

### 3.1. Sensitivity Analysis Results

In the sensitivity analysis experiment, the NEE, GPP, and leaf area index (LAI) were used as the model outputs, and the eFAST method was performed on the 19 selected sensitivity parameters (Table 2). Since the parameters with stronger sensitivity contributed most of the influence, we only selected several parameters with the highest sensitivity index for comparison.

This section includes two numerical experiments: Experiment 1 compared the changes in the sensitivity index from 2010 to 2020. The parameters with the highest sensitivity to NEE and GPP were the same (Figure 1a,b)—namely, the extinction coefficient (k), canopy average specific leaf area (SLA), leaf carbon-to-nitrogen ratio (C: N_leaf_), fraction of leaf nitrogen in RuBisCO (FLNR), and shaded-to-sunlit specific leaf area ratio (SLA_shd: sun_). The main difference between the GPP and NEE sensitivity parameters was that GPP was more sensitive to SLA than to k. NEE was much more sensitive to SLA and k than to other parameters (Figure 1c). The other two sensitive parameters of carbon flux were the new fine root carbon to new leaf carbon allocation (FRC: LC) and the new stem carbon to new leaf carbon allocation (SC: LC), which happened to be the parameters with the strongest influence on the LAI. None of the parameters showed large differences in long-term variation, and the differences between the first-order sensitivity index and the total sensitivity index were small.

On this basis, the sensitivity analysis of Experiment 2 compared the changes in different value ranges. The experiment was based on the ranges of ±10% and ±20% perturbation of the fixed parameter values in 2010. The results showed that the parameter sensitivity index of the modified parameter value range was quite different from that of the default parameter value range. It was mainly manifested in that the most sensitive parameter for the model output was C: N_leaf_, the sensitivity index of SLA dropped significantly (Figure 2), and there were also differences in the top-ranked parameters. The difference between the parameter sensitivity indices in the range of ±10% and ±20% perturbation was small; these parameters were ranked the same for different outputs, such as GPP, NEE, and LAI.

### 3.2. Model Optimization and Validation

In this study, the Biome-BGC model was used to simulate the NEE from March 2010 to February 2011, and 19 parameters—including SLA and C: N_leaf_—were optimized using the actual NEE and the PEST method. The results showed that the simulation performance of the model optimized by the PEST method was improved (Figure 3a). Before calibration, the original model had a certain degree of deviation (R^2^ = 0.53, RMSE = 1.62); it had a significant underestimate of NEE from August to October and an overestimate from March to May. The revised model was significantly improved (R^2^ = 0.72, RMSE = 1.14). The overall variation in the simulated values was reduced, and the trend was more obvious. The overall relative error of the simulation improved from 16% to 13%, but there was still a degree of underestimation.

Based on the sensitivity analysis results presented in Section 3.1, we selected several parameters with strong sensitivity from the 19 parameters—namely, SLA, k, C: N_leaf_, FLNR, SLA_shd: sun_, FRC: LC, and SC: LC—and used the PEST method to optimize only these 7 parameters. We named the scheme optimized with 19 parameters “Scheme 1” and the scheme optimized with 7 parameters “Scheme 2”. The results showed that Scheme 2 could also achieve accurate optimization results (R^2^ = 0.71, RMSE = 1.19), and the difference between the simulation results of the two optimization schemes was small (Figure 3b).

Based on the optimized model parameters, we conducted validation on the NEE from March 2011 to December 2020 (Figure 4). The simulation results showed that the model could accurately simulate NEE in other years (R^2^ = 0.64, RMSE = 0.98) and the trend of NEE in different phenological periods could be well-described. However, the overall simulation accuracy was lower than that in 2010 due to the lack of eddy covariance observation data in 2016 and other years. The overall relative error of the simulation was 12%, showing an overall underestimation of carbon sink capacity.

### 3.3. Factor Analysis of Ecosystem Carbon Flux

Based on the optimized model parameters, we simulated the interannual changes of GPP and NEE in the study area from 2010 to 2020, and then we calculated the changes in annual average temperature and total annual rainfall (Figure 5). According to the linear correlation analysis and *p*-test between the dependent and independent variables, the GPP and NEE in the study area showed a slight upward trend from 2010 to 2020, but the trend was not significant (R^2^ = 0.17, *p* > 0.05; R^2^ = 0.20, *p* > 0.05). The annual average temperature showed a large fluctuation and a slight upward trend (R^2^ = 0.15, *p* > 0.05), the annual rainfall showed a downward trend (R^2^ = 0.16, *p* > 0.05), and the trends were not significant. Pearson’s correlation analysis showed that there was a weak correlation between annual rainfall and GPP (pcc = 0.26)—both of them showed a downward trend in 2010–2015 and an upward trend in 2015–2020. From the appearance time of extreme climate, the temperature had a limiting effect on GPP. For example, 2011 and 2013 were the two years with the lowest average annual temperature, and 2015 was the year with the highest annual average temperature, all of which showed lower GPP.

## 4. Discussion

### 4.1. Ecophysiological Parameters Affecting Carbon Flux in the Rubber Plantation

The carbon flux outputs of Biome-BGC were highly sensitive to SLA, k, C: N_leaf_, FLNR, SLA_shd: sun_, FRC: LC, and SC: LC. In all of the sensitivity analysis experiments, these parameters showed strong sensitivity and were significantly higher than the other parameters (Figure 1 and Figure 2). Moreover, the multiyear sensitivity analysis results showed that the result does not change with the simulation year, so it is stable in time. Among them, SLA, k, and SLA_shd: sun_ were closely related to light. Broadleaf forests with low canopy closure tend to have higher SLA, lower k, and lower SLA_shd: sun_; thus, they receive more photosynthetic active radiation and fix more CO_2_ through photosynthesis. C: N_leaf_ and FLNR can affect the photosynthesis efficiency of leaves by controlling the contents of photosynthesis-related enzymes [33]; the processes have optimal temperatures, so they are closely related to air temperature. LAI was highly sensitive to FRC: LC and SC: LC, which affect carbon flux outputs by controlling the growth state of vegetation leaves. The carbon in the environment enters the biosphere through photosynthesis and returns to the atmosphere through respiration. These two physiological processes are crucial to the carbon cycle of the ecosystem [34]. Therefore, it is reasonable that the above parameters that can influence photosynthesis and respiration are the most sensitive parameters for the carbon flux outputs of the Biome-BGC model. We can infer that not only for the Biome-BGC model, but also for other ecosystem models, the most sensitive parameters for carbon flux outputs are still parameters related to photosynthesis and respiration. This can help with the application of the Biome-BGC model in other biomes and the optimization of other models.

Different parameter value ranges significantly affected the results of the sensitivity analysis, while the results of the analyses using the ranges of 10% and 20% perturbation of the values had no significant differences. In three different sensitivity analysis experiments, the results showed a great difference between the range of values determined by recent reports and that from the range of perturbation of values. The difference was not only in the ranking of parameter sensitivity; the latter analysis results did not show differences between carbon flux (GPP, NEE) and LAI (Figure 2). For the settings of the ranges of 10% and 20% perturbation, previous studies have shown that there are certain differences in the analysis results [20]. The reason for our undifferentiated results may be that the model was revised in this study. For the revised model, this difference can no longer be reflected in the results; it may also be because the range of 10–20% perturbation is too small to reflect the difference. In a word, the undifferentiated results are determined by the model itself. For sensitivity analysis methods, different value ranges mean different degrees of parameter variation, and the variance-based sensitivity analysis results can naturally reflect this difference [29]. The undifferentiated results show that the results of the sensitivity analysis are not only affected by the value range but also more dependent on the model itself, and this dependency has nothing to do with model selection, but for all “black-box models” that need to consider parameter values the above results can help to understand the relationship between models and parameters.

### 4.2. Applicability of the Biome-BGC Model to the Rubber Plantation in the Study Area

The Biome-BGC model is suitable for carbon flux simulation of the rubber plantation in the study area. In all experimental years, the model showed a good fit and could also accurately describe the carbon flux differences in different growth periods within a year. Unlike deciduous broad-leaved forests in temperate regions, the rubber plantations in the study area have special phenological characteristics. The general deciduous broad-leaved forests begin to grow new leaves in spring and lose them in autumn, while the rubber plantations generally drop leaves from February to March every year, and new leaves grow almost simultaneously as the previous ones drop [35]. Therefore, there is an obvious carbon flux change scenario in February and March every year. The phenological setting of Biome-BGC was determined by two parameters: the time to start growing new leaves, and the time to complete defoliation. It was found that this setting can accurately characterize carbon flux changes in the deciduous broad-leaved forests in temperate regions by adjusting parameters. However, for rubber plantations, using such a parameter setting for simulation will cause the carbon pool of the model to enter the deciduous stage without accumulating enough organic matter, leading to the model being unable to be initialized normally. Similarly, treating rubber plantations as evergreen forests makes the model unable to capture differences in carbon flux across phenological periods. In response to the above problems, our approach was to adjust the start time of the simulation; that is, by modifying the meteorological data, the time when the vegetation began to grow new leaves was used as the start time of the simulation process so that the time of the growing season was long enough for the model to initialize normally (Figure 6). This adjustment is applicable to forests whose phenological cycle cannot be described by the phenological module of Biome-BGC, so the above research helps with the application of the model in other biomes.

Parameter optimization based on observational data can effectively improve the accuracy of the model. Although the phenologically adjusted model can accurately simulate the trend of carbon flux, it is still inaccurate in describing the magnitude of carbon flux in each period, and the uncertainty of the simulation results is high (Figure 3a). The principle of the PEST method is to traverse multiple parameter groups until the parameter group with the smallest residual between the simulated and actual value is found [28], which is advantageous for parameter groups with less overall variability, so that the optimized simulation results are more stable in time. Uncertainty in model simulations arises from many aspects, including the model structure and the driving data, with model parameters being an important source of uncertainty [36]. The optimization method used in this study was based on accurate observation data. Parameter optimization targeting observation data can more effectively optimize model parameters, thereby eliminating parameter uncertainty and making the simulation results closer to reality. Furthermore, the observation data used for parameter optimization can include not just the target variable of the simulation, but also other outputs or intermediate variables, such as leaf area index [19]. The above results validate the applicability of this method in the study area and help with the application and promotion of the method in other scenarios.

### 4.3. Response of Rubber Plantation Ecosystem GPP to Meteorological Factors

Rainfall dominated the trend of GPP in the study area. Rainfall was more strongly correlated with GPP than temperature. Although the interannual trend showed that the overall trend of rainfall was decreasing, this was due to the significant decrease in 2019, while both rainfall and GPP showed a decreasing trend in 2010–2015 and an increasing trend in 2015–2020 (Figure 5a). The dominant effect of rainfall on production is manifested in several aspects. Firstly, precipitation—especially dry-season precipitation—significantly affects vegetation production. The tropics have abundant moisture and vigorous vegetation growth, while limited precipitation in the dry season inhibits the enhancement of production, highlighting its dominant role in production. Secondly, rainfall can influence plants’ root growth by altering the soil moisture content and soil nutrient flow. Previous studies have shown that the seasonal dynamics of vegetation root growth in tropical regions are consistent with soil moisture dynamics, with the rainy season tending to be the peak of vegetation growth and accumulation, while the growth is slower in the dry season [37]. Thirdly, abundant precipitation tends to occur in the same season with sufficient light and suitable temperature, promoting the growth of vegetation together and, thus, strengthening the dominant role of precipitation.

The influence of temperature was mainly manifested in the limiting effect of extreme temperatures on production. The GPP of the forest was lower in the years with the highest and the lowest mean annual temperatures (Figure 5b). According to the literature [35], the optimal temperature for the photosynthesis of rubber trees is 25–30 °C; when the ambient temperature is below 10 °C, rubber tree photosynthesis stops, while above 40 °C rubber trees’ respiration exceeds photosynthesis and growth is inhibited. The duration of high-temperature weather in the study area was long. For example, there were 88 days with maximum temperatures above 35 °C during 2015; the lowest temperatures generally occurred in December and January, with 10 days with minimum temperatures below 10 °C during 2011. Therefore, the temperature limitation of rubber plantation production in the study area mainly lies in high-temperature limitations, where the inhibited photosynthesis and enhanced respiration together reduce the accumulation of organic matter, thereby limiting the level of vegetation production. Only under chilling conditions below 10 °C did rubber plantations experience transient production limitations due to photosynthetic cessation. These findings show that higher rainfall has a long-term enhancement effect on rubber plantation production, while providing timely protection measures during extreme weather can prevent damage to rubber plantations. This helps with the management of rubber plantations in response to different climatic phenomena.

## 5. Conclusions

The conclusions of this study can be summarized as follows: (1) The carbon flux outputs of the Biome-BGC model were highly sensitive to SLA, k, C: N_leaf_, FLNR, SLA_shd: sun_, FRC: LC, and SC: LC. (2) The Biome-BGC model based on observational data and PEST was suitable for carbon flux simulation of the rubber plantation in the study area. (3) Rainfall dominated the trend of GPP in the study area, while extreme temperatures restricted GPP. This study deepens the understanding of ecological model uncertainty caused by parameter sensitivity and provides a reliable solution for the accurate simulation of carbon flux in rubber plantations, which will be helpful in future research in other regions and the sustainable development of forests. The results of the present study can be compared with the offline Community Land Model version 4.5 (CLM45) at a point scale to discuss the role of model complexity in simulating the GPP and NEE using the eFAST method.

## Figures and Tables

**Figure 1 ijerph-19-14068-f001:**
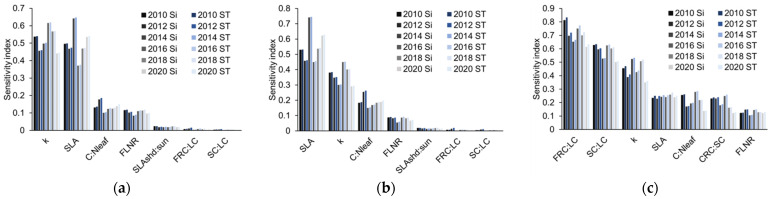
First-order sensitivity index (S_i_) and total sensitivity index (S_Ti_) of key parameters during 2010–2020 with different outputs: (**a**) with NEE as the output; (**b**) with GPP as the output; (**c**) with LAI as the output.

**Figure 2 ijerph-19-14068-f002:**
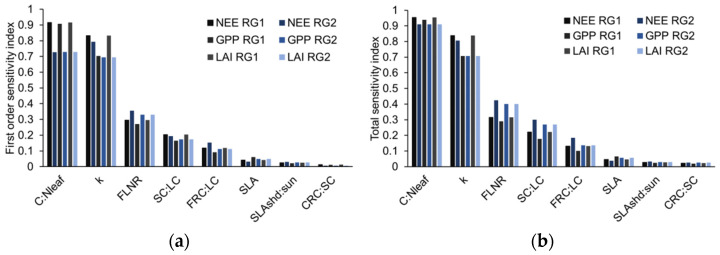
(**a**) First-order sensitivity index and (**b**) total sensitivity index in the ranges of ±10% perturbation and ±20% perturbation of parameter values.

**Figure 3 ijerph-19-14068-f003:**
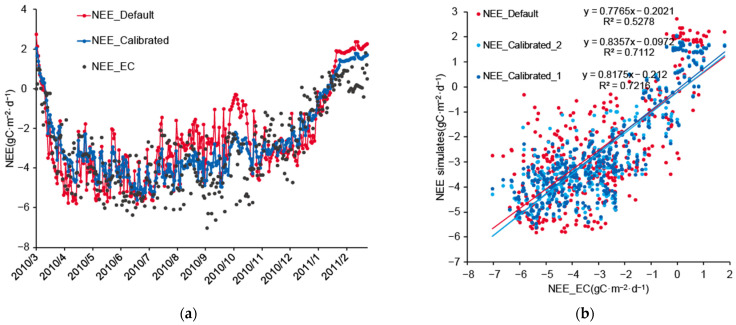
(**a**) Comparisons of NEE values simulated from the default and calibrated Biome-BGC model in 2010. (**b**) Scatterplot of the actual NEE values and the NEE values simulated from calibration with Scheme 1 and Scheme 2.

**Figure 4 ijerph-19-14068-f004:**
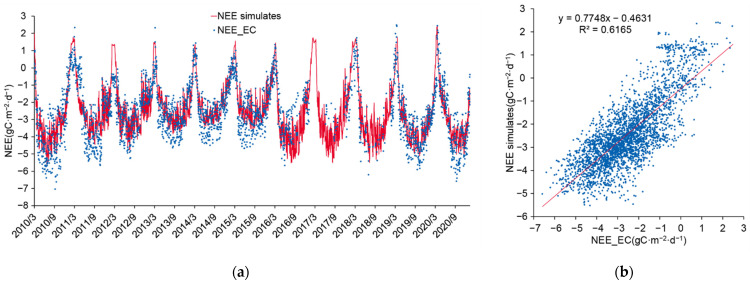
(**a**) Curve chart and (**b**) scatterplot of validation of NEE values simulated from the calibrated Biome-BGC model during 2010–2020.

**Figure 5 ijerph-19-14068-f005:**
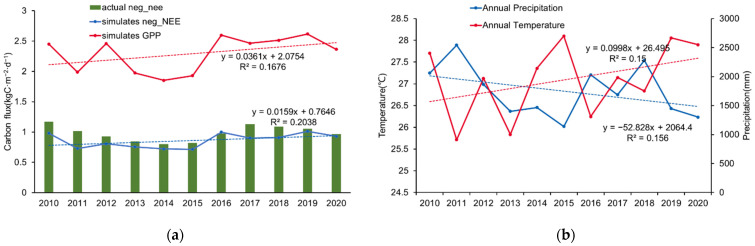
(**a**) Interannual GPP, NEE, and (**b**) meteorological data statistics in the study area during 2010–2020.

**Figure 6 ijerph-19-14068-f006:**
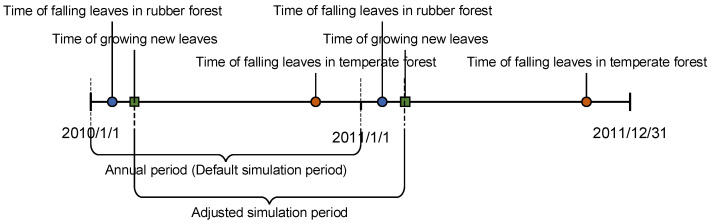
Phenology of rubber plantations and temperate deciduous broad-leaved forests, and the adjustments to simulation periods.

**Table 1 ijerph-19-14068-t001:** Site characteristics.

Parameters	Unit	Value
Effective soil depth	cm	100
Soil silt percentage	%	19
Soil sand percentage	%	52
Soil clay percentage	%	29
Elevation	m	144
Latitude	degree	19.51
Shortwave albedo	-	0.2
CO_2_ concentration	ppm	407.8
Atmospheric nitrogen deposition	g N m^−2^ a^−1^	1.71

**Table 2 ijerph-19-14068-t002:** Definitions, abbreviations, ranges, and sources of the 19 parameters of the Biome-BGC model used in the sensitivity analysis.

Parameters	Abbreviation	Range	Reference
Annual leaf and fine root turnover fraction	LFRT	[0.5, 0.828]	[30]
Annual live wood turnover fraction	LWT	[0.56, 0.9]	[18]
New fine root C: new leaf C	FRC: LC	[0.545, 1.59]	[18]
New stem C: new leaf C	SC: LC	[0.84, 1.56]	[18]
New live wood C: new total wood C	LWC: TWC	[0.096, 0.279]	[18]
New root C: new stem C	CRC: SC	[0.077, 0.563]	[18]
C: N of leaves	C: N_leaf_	[8.96, 13.44]	Measured
C: N of leaf litter	C: N_lit_	[32.88, 49.32]	[11]
C: N of fine roots	C: N_fr_	[37.92, 56.88]	[11]
C: N of dead wood	C: N_dw_	[240, 360]	[11]
Canopy water interception coefficient	W_int_	[0.0328, 0.0492]	[31]
Canopy light extinction coefficient	k	[0.56, 0.8]	[11]
All-sided–projected leaf area ratio	LAI_all:proj_	[1.71, 2.29]	[18]
Canopy average specific leaf area	SLA	[13, 26.4]	[18]
Ratio of shaded SLA: sunlit SLA	SLA_shd: sun_	[1.6, 2.2]	[18]
Fraction of leaf N in RuBisCO	FLNR	[0.048, 0.072]	[11]
Maximum stomatal conductance	g_smax_	[0.004, 0.006]	[32]
Leaf water potential: complete	LWP_f_	[−3.9, −1.5]	[18]
Vapor pressure deficit: complete	VPD_f_	[2300, 4700]	[18]

## Data Availability

Not applicable.

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
