# Peer review of "Sensitivity Analysis of Biome-BGC for Gross Primary Production of a Rubber Plantation Ecosystem: A Case Study of Hainan Island, China"

_ijerph, 2022, doi:10.3390/ijerph192114068_

Round 1

Reviewer 1 Report

Dear Authors,

Good evening. I would like to congratulate you for your efforts in the field of biogeochemistry particularly using an efficient method to optimize the model parameters using Fluxnet observations. I recommend acceptance of the submitted paper after considering some notes in the attached file.

Congratulations for your nice work.

Author Response

Dear Reviewer:

On behalf of my co-authors, we thank you very much for giving us an opportunity to revise our manuscript, we appreciate your positive and constructive comments and suggestions on our manuscript entitled “Sensitivity analysis of Biome-BGC for gross primary production of rubber plantation ecosystem: A Case Study of Hainan Island, China”. (ID: ijerph-1900230).

We have studied your comments carefully and have made revisions which are marked in red in the response file. We have tried our best to revise our manuscript according to the comments. Attached please find the response file, which we would like to submit for your kind consideration.

Thank you and best regards.

Yours sincerely,

Junyi Liu

Name: Zhixiang Wu

Reviewer 2 Report

Comments on “Sensitivity analysis of Biome-BGC for gross primary production of rubber plantation ecosystem: A Case Study of Hainan Island, China” submitted by Junyi Liu et al.

In this study, the authors applied extended Fourier Amplitude Sensitivity Test (eFAST) method to investigate the most sensitive ecophysiological parameters of the Biome-BioGeochemical Cycles (Biome-BGC) model, and then used the flux data and the Model-independent Parameter Estimation (PEST) method to simulated the carbon flux of the rubber plantation ecosystem in the study area from 2010 to 2020. The manuscript is very well written, clear and fluent. Overall purpose of the research is sound and of high importance in understanding the parameter sensitivity on carbon cycle. The authors provide most of the information needed to understand the model, the outcome and the interpretation of the results and once the major issues are addressed it could be recommended for publication.

Comment1: Is the range data in Table 2 reasonable? for example, the maximum value of FRC:LC data sometimes exceed 2, especially in dry season. A wider range would make the simulations design more clearly and understandable.

Comment2: Relationships between GPP/NEE and climate factors are not well illustrated. Are they positively related? I would suggest the authors to analyze their relationship using mathematical methods, which would make this section more convincible.

Comment 3: I recommend you discuss all the parameters although you only choose 19 of them. For example, “year day to start new growth (when phenology flag = 0)” and “transfer growth period as fraction of growing season” also have considerable impact.

Specific comments:

1.      Line 62: “because it relies only on traditional meteorological data to drive” doesn't seem like a reasonable English expression.

2.      Line 388, line 392: the expression should be secondly and thirdly? Please check.

3.      Line 410: “The study shows that a stable source of water has a long-term enhancement effect on rubber plantation” how did you get to this conclusion?

Author Response

(The authors gave the same response as above.)

Round 2

Reviewer 2 Report

The paper can in principle be accepted after revision.